# *Pereskia aculeata* Miller as a Novel Food Source: A Review

**DOI:** 10.3390/foods12112092

**Published:** 2023-05-23

**Authors:** Naaman Francisco Nogueira Silva, Sérgio Henrique Silva, Daniel Baron, Isabelle Cristina Oliveira Neves, Federico Casanova

**Affiliations:** 1Centro de Ciências da Natureza, Universidade Federal de São Carlos (UFSCar), Buri 18290-000, SP, Brazil; danielbaron@ufscar.br (D.B.); isabelle@ufscar.br (I.C.O.N.); 2Instituto de Ciências Exatas e Tecnológicas, Universidade Federal do Triângulo Mineiro (UFTM), Uberaba 38025-180, MG, Brazil; sergio.henrique@uftm.edu.br; 3Food Production Engineering Group, DTU Food, Technical University of Denmark, Søltofts Plads 227, Dk-2800 Lyngby, Denmark

**Keywords:** plant proteins, mucilage, bioactive molecules, sustainable protein sources, functional properties, ora-pro-nobis

## Abstract

*Pereskia aculeata* Miller is an edible plant species belonging to the Cactaceae family. It has the potential to be used in the food and pharmaceutical industries due to its nutritional characteristics, bioactive compounds, and mucilage content. *Pereskia aculeata* Miller is native to the Neotropical region, where it is traditionally employed as food in rural communities, being popularly known as ‘ora-pro-nobis’ (OPN) or the Barbados gooseberry. The leaves of OPN are distinguished by their nontoxicity and nutritional richness, including, on a dry basis, 23% proteins, 31% carbohydrates, 14% minerals, 8% lipids, and 4% soluble dietary fibers, besides vitamins A, C, and E, and phenolic, carotenoid, and flavonoid compounds. The OPN leaves and fruits also contain mucilage composed of arabinogalactan biopolymer that presents technofunctional properties such as thickener, gelling, and emulsifier agent. Moreover, OPN is generally used for pharmacological purposes in Brazilian folk medicine, which has been attributed to its bioactive molecules with metabolic, anti-inflammatory, antioxidant, and antimicrobial properties. Therefore, in the face of the growing research and industrial interests in OPN as a novel food source, the present work reviews its botanical, nutritional, bioactive, and technofunctional properties, which are relevant for the development of healthy and innovative food products and ingredients.

## 1. Introduction

The continuous increase in the global population associated with agricultural expansion, climate changes, and awareness of the importance of ecological preservation has pressured researchers and industries to find more sustainable food sources [1,2,3]. In fact, animal source foods are related to increments of deforestation, greenhouse gas emissions, water consumption, and risks to human health [4,5]. As a consequence, there is a growing interest in plant-source foods as an answer to more sustainable agricultural practices [2,6]. Moreover, food consumption has also been influenced by ethnic, cultural, and religious beliefs, with an increasing number of people with dietary restrictions, such as vegans, vegetarians, and flexitarians. In this way, plant-source foods can satisfy the demand for edible proteins while providing essential nutrients to the human diet in a more sustainable manner [3,7].

Recently, many unconventional edible plants have been researched regarding their nutritional, biological, and industrial potentialities [8,9]. Among these alternative plant species, *Pereskia aculeata* Miller is a nutrient-rich cactus native to Latin American countries [10] that offers interesting characteristics for food and pharmacological applications. *P. aculeata* Miller is also known as ‘ora-pro-nobis’ (OPN), blade-apple cactus, Barbados gooseberry, leaf cactus, lemon vine, and rose cactus. The leaves of OPN are distinguished by their nontoxicity and nutritional richness, including, on a dry basis, 23% proteins, 31% carbohydrates, 14% minerals, 8% lipids, and 4% soluble dietary fibers, besides significant content of vitamins A, C, and E and diverse phenolic, carotenoid and flavonoid compounds [11,12,13,14]. Furthermore, the OPN leaves and fruits contain mucilage composed of arabinogalactan biopolymer that presents technofunctional properties such as thickener, gelling agent, and emulsifier [15,16,17], as well as wound healing characteristics [18]. These assets demonstrate the potential use of OPN mucilage in the food and pharmaceutical industries.

In this context, this review focuses on the general aspects of OPN as a novel food source, including the botanical, nutritional, bioactive, and technofunctional properties of OPN leaves and fruits. We evaluated 241 scientific articles available on Web of Science and 34 on Scielo (Latin American scientific database) that contained the word “*Pereskia*” in any part of the text. It should be highlighted that this review was inspired by the potential of using OPN for the development of healthy, affordable, and innovative food products and ingredients, particularly in the case of Latin American countries where OPN is naturally found.

## 2. General Botanical Characteristics

According to an update of the angiosperm phylogeny group (APG) classification [19], the Cactaceae family is circumscribed to the eudicots group (or tricolpate), Superasterids clade, Caryophyllales order. Indeed, Cactaceae is one of the most interesting angiosperm botanical families, as it presents a photosynthetic mechanism that concentrates atmospheric inorganic carbon, described as crassulacean acid metabolism (CAM), specialized for survival in arid environments. CAM allows the opening of the stomata complex at night (saving water) to capture carbon dioxide (CO_2_) that is stored in the form of malic acid (or malate) and used in photosynthesis during the next day [20]. Such a physiological mechanism permits their survival during prolonged drought while keeping a healthy tissue water status [21]. The Cactaceae family includes about 100 genera and approximately 1500 species that are mainly distributed in the Neotropical region (Mexico, Brazil, and Chile) except for the *Rhipsalis* genus that occurs in tropical Africa, besides many species of the *Opuntia* genus introduced in Africa, Australia, and India [22]. Among the main genera belonging to the Cactaceae family, *Pereskia* genus (subfamily Pereskioideae) is a leafy shrub and tree native to the American continent, comprising 17 species that are distributed from Argentina to Florida [10,21,23].

Phylogenetic analysis reveals that *Pereskia* is a paraphyletic genus that originated from all other cactus species. *Pereskia* retained numerous plesiomorphic character states, such as non-succulent stems, persistent leaves, cymose inflorescences, stylet (polyestemon), and, in some species, a superior ovary with basal placentation [22]. As these characteristics are absent in other Cactaceae family members, it is reported that *Pereskia* originated a clade characterized by the presence of complex stomata in the stem and by the late formation of the bark, which promoted photosynthesis in the stem [22]. In this sense, the *Pereskia* genus is considered an ancestral of the modern cacti. Although the species belonging to the *Pereskia* genus can grow under limited water availability (as a conventional cactus does), they differ from the other cacti mainly because they do not possess a photosynthetic stem and also because they present well-developed leaves, in which water and most nutrients are accumulated [14,21,24,25]. Among the *Pereskia* species, *Pereskia aculeata* Miller (OPN), in particular, is traditionally employed as food in Brazilian rural communities due to the sensorial acceptance and high nutritional content of its leaves, including significant amounts of proteins, fibers, minerals, and vitamins [14,26,27].

Regarding the OPN morphological characteristics, it can be described as a liana (woody vine) perennial and shrubby species (Figure 1), reaching 4 m in height or even more, that has become an invasive plant in the world [28]. The older well-lignified stems are endowed with prominent spines that demand care for handling [23,29]. It presents long branches and simple elliptical leaves with short petioles and succulent texture, from 3–8 cm up to 15 cm long [23,29]. The inflorescences are short, numerous, with cream-yellow flowers, sometimes with a red center, arranged on the foliage and formed in spring or summer, visited by bees and carpenter bees. The OPN fruits, when ripe, are globular, yellowish, edible, and berry-like, with glochids and black seeds [30] (Figure 2).

With respect to agronomic performance, OPN is a rustic species, tolerant to arid conditions, that presents vegetative growth all year long and that does not require high soil fertility; consequently, it is adapted to diverse soil types and edaphoclimatic conditions [23,29]. This species can be propagated with stem cuttings or by seeds and can also be used as a rootstock for other cacti [31]. When cultivated at high density (~10 plants per m²), OPN can produce 5759 kg of proteins in the leaves (main edible part) per hectare per year [24]. For comparison purposes, considering only the grains, the average soybean (*Glicyne max* (L.) Merr.) and maize (*Zea mays* L.) protein productions per hectare per year are 1154 kg and 514.7 kg, respectively [24].

## 3. Nutritional Characteristics of OPN Leaves

Several works have studied the chemical composition of OPN leaves, and the results are summarized in Table 1. As can be seen (Table 1), almost 90% *w*/*w* of OPN leaves are represented by water. The remaining dry matter is quite diverse and contains all classes of essential nutrients to the human diet.

Carbohydrates represent approximately one-third of the dry matter of OPN leaves. They are present as structural and highly ramified polysaccharides formed by galactopyranose, arabinofuranose, arabinopyranose, rhamnopyranose, uronic acid, and fucose [16,35]. These complex polysaccharides are known as mucilage (non-toxic) and can be used as hydrocolloids in food processing due to their high water absorption capacity [12,36,37]. In fact, the first process for obtaining OPN mucilage was described in 1982 in Sierakowski’s protocol [37] by using benzene, ethanol, water, and acetone. This study showed that OPN mucilage was composed of an arabinogalactan, formed mainly by arabinose and galactose, containing 3.5% protein in relation to the polysaccharide content. Lima-Junior et al. [35] optimized the OPN mucilage extraction process by using only ethanol as a solvent, and the obtained mucilage (as powder) presented 10.5% protein and 46.9% carbohydrates. OPN leaves have a high arabinogalactan content, characterized as a biopolymer primarily composed of a main backbone of (1→4) β-D-galactopyranose with branches of galactose, arabinose, rhamnose, and galacturonic acid at the C-3 position, in the proportion of 5.4:8.3:1.8:1.0, respectively [16,35]. Size exclusion chromatography analyses revealed that the arabinogalactan material is heterogeneous, presenting an average molar mass of 7.9 × 10^5^ g/mol [16]. This arabinogalactan is considered ‘type I’ because proteins are associated with the polysaccharide chain by covalent bonds. This proteoglycan complex is typical of the cell walls of higher plants, and it is related to the physical–chemical properties of OPN mucilage [12,16].

Dietary fiber is a plant material resistant to enzymatic digestion, which is essential to human health because its consumption has been associated with a decreased incidence of numerous diseases [38]. Thus, another positive nutritional aspect of OPN leaves is their significant content of dietary fibers, including both soluble and insoluble fractions. Concerning the lipid fraction, which is also present in a considerable concentration in OPN leaves, to date, no work has evaluated their composition, i.e., the already published papers were limited to analyzing the total lipid concentration by Soxhlet methods [13,32,33,39]. Concerning minerals and vitamins, their contents in OPN leaves are impressive. Considering the nutrient reference value requirements (per day) for an adult, established by FAO [40], a portion of 30 g of OPN leaf flour (dried leaves) can provide, on average: 100% calcium, 100% magnesium, 160% phosphorus, 250–400% iron, 230–290% zinc, 430% copper, 3200% manganese, 900% vitamin A, 340% vitamin C, and 100% vitamin E.

Regarding the protein fraction, it is interesting to note that the protein content of OPN leaves, on a dry basis, is quite similar to whole milk powder from cows (~25% *w*/*w*) [41], which underlines the nutritional relevance of OPN leaves for human consumption. In this sense, their amino acid composition was analyzed by Takeiti et al., (2009) [13] and Silveira et al., (2020) [42]. Although these works diverged about the amino acid proportions, both reported that OPN leaves provide all essential amino acids in suitable proportions for child and adult nutrition. Only methionine and lysine were present slightly below the ideal levels recommended by the World Health Organization [43]. Furthermore, the protein fraction must be digestible by the human gastrointestinal tract to be considered a good protein source. According to Takeiti et al., (2009) [13], the in vitro digestibility of OPN leaf proteins was 75.9%, which is similar to the digestibility of beans, wheat, and rice [43].

Considering the physical–chemical characteristics, it was demonstrated by sodium dodecyl sulfate–polyacrylamide gel electrophoresis (SDS-PAGE) that the OPN leaf proteins present molar masses ranging from 15 to 97 kDa, with significant bands found at 61 kDa, 53 kDa, 33 kDa, and 15 kDa [13]. After separation by cryogel chromatography, followed by Fourier-transform infrared (FTIR) spectroscopy, Neves et al., (2020) [44] showed that the secondary structures of OPN leaf proteins were mainly composed of β-sheet (46.5%) and α-helix (13.9%), indicating a stable conformation for these proteins. They also observed a zeta potential of 5.8 mV at pH 3.2, which suggests an isoelectric point slightly above this pH value. Finally, Morais et al., (2019) empirically evaluated the protein recovery of OPN leaf proteins by combining salting out, temperature, and isoelectric precipitation [45]. These authors noticed that the highest protein recovery in the presence of precipitating salts (about 69%) was achieved using ammonium sulfate (NH_2_SO_4_) at 0.5 M and 85 °C. In the absence of precipitating salts, the highest protein recovery was found between pH 3 (71%) and pH 4 (69%) at 85 °C. These results agree with Neves et al., (2020) [44] for an isoelectric point of OPN leaf proteins between pH 3 and 4. Ultimately, by accessing all scientific papers containing the word “*Pereskia*” in any part of the text (by using Web of Knowledge and Scielo databases), it was verified that only three scientific works—Takeiti et al., (2009) [13], Morais et al., (2019) [45] and Neves et al., (2020) [44]—presented qualitative physical–chemical information about OPN leaf proteins.

## 4. Bioactive Properties of OPN Leaves

OPN is also popularly used for pharmacological purposes in Brazilian folk medicine, which is attributed to its metabolic, anti-inflammatory, antioxidant, and antimicrobial properties, among others, as reviewed by Agostini-Costa (2020) [46] and Porto et al., (2021) [14]. Concerning the metabolic aspects, Barbalho et al., (2016) [39] administrated OPN leaf flour to Wistar rats and observed significant health improvements, including a reduction in weight gain, visceral fat, levels of total cholesterol, triglycerides, low-density lipoprotein, very-low-density lipoprotein, and increased HDL-c and enhancement of intestinal motility. Moreover, in double-blind, randomized clinical studies with humans, Vieira et al., (2019, 2020) [47,48] noticed that the consumption of OPN leaf flour in biscuits and beverages improved intestinal health and reduced weight, waist circumference, and body fat and increased satiety. The authors suggested that these results are associated with the presence of dietary fiber and phytochemical compounds, such as polyphenols, in the OPN leaves.

Another important pharmacological use of OPN is related to its anti-inflammatory properties. Pinto et al., (2015) [49] studied the topical anti-inflammatory activity of a hexane extract of OPN leaves in acute and chronic ear dermatitis in mice. The authors verified that the OPN extracts significantly reduced inflammatory processes induced by various toxic agents. In addition, the OPN extracts did not exhibit toxicity for dermatological applications. The same research group [50] also verified an antinociceptive activity using a methanolic OPN extract in mice treated with acetic acid. The authors attributed the analgesic effect to alkaloids and quercetin in OPN extract. Carvalho et al., (2014) [18] produced ethanolic extracts of OPN cultivated in different soil types and evaluated their in vitro wound healing properties and cytotoxicity using mouse fibroblast cells. They observed that OPN extracts obtained from all tested soils were safe and effective as a wound healing agent, whose beneficial effects were attributed to the mucilage of the OPN leaves. The main health-related effects of OPN leaves are summarized in Table 2.

One of the main popular claims of OPN leaves concerns their antioxidant properties. In fact, OPN leaves contain high concentrations of several classes of antioxidants, including carotenoids (α and β-carotene, lutein, zeaxanthin, and violaxanthin) [51], phenolic compounds such as caffeic, chicoric, and coumaric acid derivatives, flavonoids (quercetin, kaempferol, and isorhamnetin glycoside derivatives) [52,53], and terpenoids (phytol, γ-tocopherol, vitamin E, squalene, and lupeol) [54], among others. In general, the extraction procedures of antioxidant molecules from the OPN leaves are achieved by mixing them with organic solvents, such as ethanol, acetone, methanol, and hexane [51,52,53]. However, it is also possible to improve antioxidant extraction using supercritical fluid technology with CO_2_ [54]. In a study focused on the antioxidant properties of OPN leaves [51], the authors produced a hydroethanolic extract that was effective in inhibiting the growth of Gram-positive and Gram-negative bacteria, including human pathogens such as *Listeria monocytogenes*, *Staphylococcus aureus*, and *Klebsiella pneumoniae*. Last, but not least, it is noteworthy that the works evaluating the toxicity of OPN leaves to humans showed that they are safe for food and therapeutic applications [52,55].

## 5. Technofunctional Properties of OPN Leaves

From a food processing perspective, technofunctional properties or technological functionalities can be defined as the effectiveness with which a particular food component or ingredient provides desirable attributes in a food or beverage [8,56]. To date, the technofunctional properties of OPN leaves have been studied in the formulation of food products in the forms of flour [48,57,58], mucilage [17,36], and fresh leaves [59].

In terms of flour, e.g., dried and grounded OPN leaves, Sobrinho et al., (2015) [57] evaluated the effects of OPN leaf flour on the texture, color, and sensory acceptance of cooked sausages. The authors observed that sausages with 1–2% of OPN leaf flour were darker and softer than the control ones, while sensory acceptance was kept unchanged. In this same direction, Sato et al., (2018) [58] studied the impact of adding 10–20% of OPN leaf flour on the functional and sensory characteristics of pasta. The authors noticed that, compared with the control, the enriched samples were darker, softer, gained more weight during cooking, lost less weight after cooking, and presented higher contents of proteins, fibers, and minerals. At the same time, the color, flavor, odor, texture, and overall appearance were equally appreciated by the consumers of both types of pasta. Finally, in a double-blind, randomized study, Vieira et al., (2020) [48] researched the effect of a dairy-based beverage containing 5% *w*/*w* OPN leaf flour on intestinal microbiota, gastrointestinal symptoms, and anthropometric parameters in women aged from 20 to 60 years old. The results demonstrated that the daily consumption of the beverage with OPN leaf flour for 6 weeks improved feces consistency, increased satiety, and reduced weight, waist circumference, and percent body fat. However, no effect was detected on the composition of the intestinal microbiota. Briefly, these works show that it is possible to use OPN leaf flour to produce various food products, improving their nutritional content while maintaining their technofunctional properties and sensory acceptance.

Recent studies have demonstrated the potential of OPN mucilage for industrial applications. It was shown that OPN mucilage could form gels and emulsions at different hydrocolloid concentrations and temperatures [35,60,61] in the presence or not of sodium chloride and sucrose and at different pH values [62]. Furthermore, OPN mucilage is able to form oil-in-water nanoemulsions through ultrasonication [63] and can be used to produce biodegradable films when associated with glycerol [64]. The technofunctional properties of OPN mucilage were also studied in fermented dairy drinks, leading to increased protein content and viscosity of the product, as well as reduced syneresis [65]. Neves et al., (2020) [66] produced microparticles of OPN mucilage and whey protein isolate to encapsulate α-tocopherol loaded in canola oil or coconut oil. After encapsulation, the bioactive activity of α-tocopherol was retained over 35 days, and its bioaccessibility was higher when using canola oil as a carrier. Lise et al., (2021) [36] analyzed the potential of OPN mucilage as an emulsifier and fat replacer in producing a mortadella-type meat product. The authors verified that it was possible to reduce the fat content of mortadella without impairing its texture and sensory characteristics.

Additionally, even as fresh leaves, OPN is technologically viable in food formulations, such as chocolate cake production, with a consequent increase in protein, fiber, and mineral contents and a reduction in total calories [59]. Neves et al., (2021) [67] evaluated the bioactive compound levels, antioxidant activity, and bioaccessibility of carotenoids from OPN leaves submitted to different cooking methods. It was found that the cooking techniques (stir-frying, microwaving, and steaming) influenced the phytochemical composition of OPN leaves. In general, the cooking processes softened plant tissues, increased bioactive compounds’ extraction, and enhanced their bioaccessibility. These results may be used to improve dietary intake recommendations for bioactive compounds and increase the nutritional value of food products.

Overall, the available data highlight the technofunctional and nutritional potential of using OPN leaves in food processing, whether as dried/fresh leaves or isolated extract (mucilage). However, to date, no paper has studied the effects of isolated OPN leaf proteins on the gelling, emulsifying, and foaming properties of food products. This shows the vast scientific potential to study these proteins and develop new food products from OPN leaves.

## 6. Characteristics of OPN Fruit

The OPN fruit originates from its flower (monocline and actinomorphic), where the ovary occurs within the hypanthium, being considered, therefore, as superior [68]. The flowers are arranged in terminal panicles, composed of four ovate green petals measuring 0.6 to 0.8 cm in length, and the corolla has 8 to 12 oblong white petals (longer than wide) of white color measuring 2 cm in length. This hypanthium presents a green color, fleshy consistency, thick cell walls, and green lanceolate bracteoles, besides spines at the bases of the leaves. At the beginning of its development, OPN fruit consists of a succulent hypanthium, pericarp, and seeds wrapped in gelatinous material [69]. In the young pericarp is observed a differentiation in mucilaginous cells, which occurs in large numbers in the innermost region of the mesocarp [69]. With the ripening process, the hypanthium acquires a yellow-orange color (Figure 2B) while it loses its bracteoles and aculeus; during this period, it develops a small opening in the apical region [68,69]. In the mature stage, the fruits are rounded, dark yellow, with a yellowish and sweet pulp involving two to four discoid seeds of brownish-black color. The inner part of the mesocarp and the entire endocarp change their structure, forming a gelatinous mass that surrounds the seeds [69,70].

Adult OPN plants produce their fruits between June and July, which are edible, pomaceous, cactidium-type, and small (1.8 to 2.0 g per fruit) [70]. The chemical composition of OPN ripe fruits was studied by Queiroz et al., (2009) [71] and Agostini-Costa et al., (2012) [70]. The average chemical content of OPN fresh fruits is shown in Table 3. In addition, Agostini-Costa et al., (2012) [70] characterized the total and phenolic carotenoid profile of mature OPN fruit, finding trans-β-carotene as the primary carotenoid, followed by α-carotene, lutein, and other minor carotenoids. Unlike the leaves, the use of OPN fruits for human nutrition is still very restricted due to the lack of studies on their properties and the difficulties in collecting and processing it. Consequently, research into the nutritional properties of OPN fruits can broaden its use, both in natura and after processing, as jam, syrup, or juices, for example.

As observed for OPN leaves, the green fruits of OPN present mucilage with thickening and emulsifying properties. Silva et al., (2019) [72] extracted mucilage from OPN green fruits (Figure 2A) by cold extraction and lyophilization processes and investigated its physicochemical properties. The lyophilized mucilage from OPN green fruit exhibited, on a dry basis: 2.9% moisture, 67.2% carbohydrates, 19.8% proteins, 8.3% ashes, 0% lipids, 4.6% crude fiber, 0.2% phosphorus, 1.8% potassium, 2.7% calcium, 0.07% magnesium, 0.2% sulfur, 0.001% boron, 0.002% copper, 0.004% manganese, 0.005% zinc, 0.04% iron, and 0.02% sodium. FTIR spectroscopy of the mucilage obtained from OPN green fruit showed characteristic bands of polysaccharides and protein chains and an additional peak in 1722 cm^−1^. This peak referred to the presence of carboxylic groups that serves as chemical sites for ionic bonds, which contribute to the ability to form gels [72,73]. Moreover, solutions of the mucilage from OPN green fruit with increasing concentrations (0.5, 1.0, 1.5, 2.0, and 2.5 g/100 mL of aqueous phase) presented pseudoplastic behavior [72]. It was also observed that increments of the mucilage concentration increased the apparent viscosity, emulsification capacity, and emulsion stability and decreased the mean size of the oil droplets.

## 7. Conclusions

OPN is a nonconventional edible plant adapted to diverse climate conditions that requires low soil fertility for satisfactory vegetative growth. As OPN is a cactus species that does not possess a photosynthetic stem, most of the water and nutrients are accumulated in its leaves, which are employed in traditional South American dishes. The OPN leaves are nontoxic and present all nutrient classes in significant amounts, including proteins, carbohydrates, fibers, lipids, minerals, vitamins, and diverse phenolic, carotenoid, and flavonoid compounds with metabolic, anti-inflammatory, antioxidant, and antimicrobial activity. Moreover, OPN leaves and fruits have been identified as an interesting source of mucilage composed of polysaccharides and proteins. This mucilage has a heterogeneous macromolecular profile with a polyelectrolyte behavior that can be employed as an emulsifying and stabilizing agent due to its interfacial adsorption properties. Regarding the protein fraction, the OPN leaf proteins are still little studied. However, when cultivated at high density, the protein productivity per hectare can be 500% higher than that of soybeans (*G. max*) and 1,100 % higher than that of maize (*Z. mays*), demonstrating the tremendous potential of OPN leaves as a perennial source of nutrients for human health and ingredients for the food industry. In general, the scientific literature demonstrates that it is possible to use OPN leaves in the form of flour, fresh leaves, or isolated extract (e.g., mucilage) to improve the nutritional content of food products while keeping their technofunctional properties. At the same time, there is still a broad field for future research and development focused on food applications of OPN leaves and fruits, comprising the optimization of their components’ extraction as well as their use as gelling, emulsifying, and foaming agents in food formulation.

## Figures and Tables

**Figure 1 foods-12-02092-f001:**
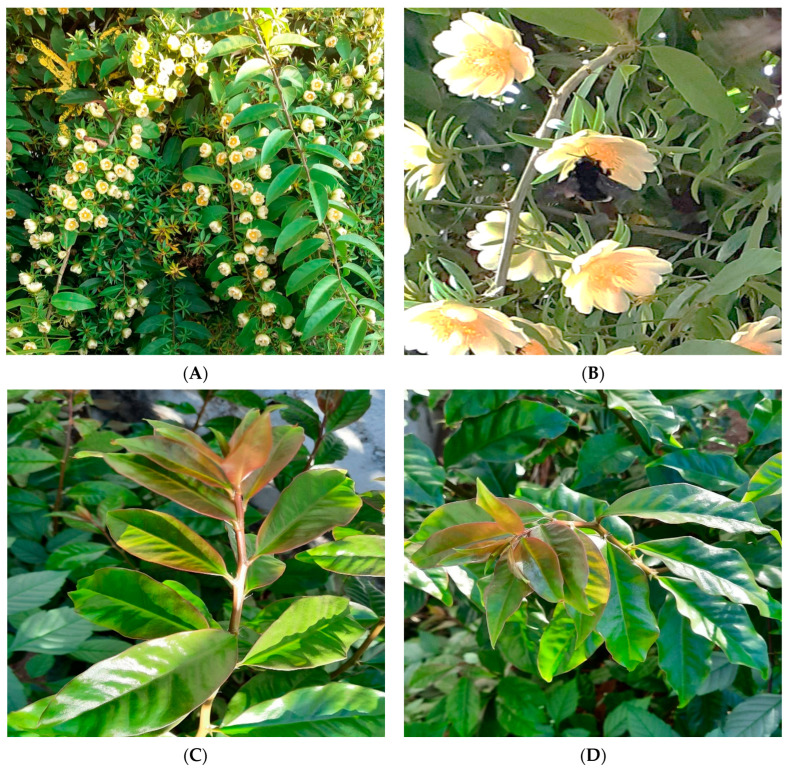
(**A**,**B**) Flowers and (**C**,**D**) leaves of *Pereskia aculeata* Mill.

**Figure 2 foods-12-02092-f002:**
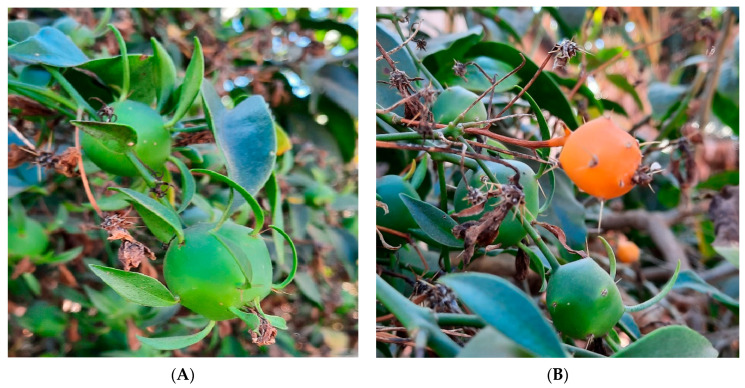
(**A**) Green fruit and (**B**) mature fruit of *Pereskia aculeata* Mill.

**Table 1 foods-12-02092-t001:** Nutritional composition of *Pereskia aculeata* Mill. (ora-pro-nobis) leaves. The values are presented on a dry matter basis.

Nutrient	Range	Average	Reference
Moisture fresh leaves (% *w*/*w*)	83.3–91.1	88.1	[13,24,32,33]
Proteins (% *w*/*w*)	14.3–29.0	23.3	[13,24,32,33,34]
Lipids (% *w*/*w*)	4.1–16.3	8.5	[13,32,33]
Carbohydrates (% *w*/*w*)	29.5–32.3	30.9	[32,33]
Soluble dietary fiber (% *w*/*w*)	2.4–5.2	3.8	[13,32]
Insoluble dietary fiber (% *w*/*w*)	19.2–33.9	26.6	[13,32]
Minerals (% *w*/*w*)	10.8–16.1	13.9	[13,32,33]
Calcium (% *w*/*w*)	1.3–4.6	3.4	[13,24,32,33]
Magnesium (% *w*/*w*)	0.6–1.9	1.0	[13,24,32,33]
Potassium (% *w*/*w*)	1.6–3.9	3.1	[13,24,32]
Phosphorous (mg/kg)	1560–5600	3770	[13,24,32,34]
Manganese (mg/kg)	88–464	327	[13,24,32]
Zinc (mg/kg)	37–267	106	[13,24,32,34]
Iron (mg/kg)	142–244	190	[13,24,32,34]
Copper (mg/kg)	12–14	13	[13,32]
Vitamin E (mg/kg)	14–49	32	[33,34]
Total carotenoids (mg/kg)	250–354	302	[33,34]
β-carotene (mg/kg)	78–430	250	[13,32,33,34]
Vitamin A (mg/kg)	23–25	24	[13,33]
Vitamin C (mg/kg)	430–1858	1144	[13,32]

**Table 2 foods-12-02092-t002:** Summary of the main health-related properties of OPN leaves.

Properties	Type of Study	Main Results	References
Metabolic	In vivo using Wistar rats	Improved metabolic profile Increased intestinal mobility	[39]
	In vivo randomized cross-over intervention with adult men	Improved gastrointestinal symptoms Increased satiety	[47]
	In vivo double-blinded randomized clinical trial with adult women	Reduced weight, waist circumference, body fat, eructation, and constipation Increased satiety Improved feces consistency	[48]
Anti-inflammatory	In vivo using Swiss and Wistar rats	Reduction of the inflammatory process of acute and chronic ear dermatitis Absence of dermal toxicity	[49]
	In vivo using Swiss rats	Antinociceptive activity (analgesic effect)	[50]
	In vitro using fibroblast cells L929	Wound healing capacity Absence of cytotoxicity	[18]

**Table 3 foods-12-02092-t003:** Chemical composition of OPN ripe fruits.

Characteristic	Range	Average	Reference
Moisture (% *w*/*w*)	87.4	87.4	[71]
Ashes (% *w*/*w*)	0.9	0.9	[71]
Protein (% *w*/*w*)	0.0–1.0	0.5	[70,71]
Lipids (% *w*/*w*)	0.2–0.7	0.5	[70,71]
Carbohydrates (% *w*/*w*)	6.3–11.5	8.9	[70,71]
Ascorbic acid (% *w*/*w*)	2.0–125.0	63.5	[70]
Niacin (mg/100 g)	0.9	0.9	[70]
Calcium (mg/100 g)	174.0–206.0	190.0	[70]
Phosphorus (mg/100 g)	26.0	26.0	[70]
pH at 22 °C	4.2	4.2	[71]

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
