# Peer review of "Pereskia aculeata Miller as a Novel Food Source: A Review"

_foods, 2023, doi:10.3390/foods12112092_

Round 1
Reviewer 1 Report
The present study is associated with the potentiality of Pereskia aculeata Miller to be used as a novel food source due to its rich composition and nontoxicity.
Abstract part is sufficient and well written.
4. Bioactive properties OPN leaves: I suggest the authors to construct a table for this section and give some details about the studies performed.
6. Characteristics of OPN fruit: This part also seems extremely complicated, and the literature studied should be better tabulated with the citations.
The conclusion part is well written, and, in the end, the future studies are directed.
Author Response
The present study is associated with the potentiality of Pereskia aculeata Miller to be used as a novel food source due to its rich composition and nontoxicity.
Abstract part is sufficient and well written.
- Bioactive properties OPN leaves: I suggest the authors to construct a table for this section and give some details about the studies performed.
This is a pertinent recommendation, and we inserted a new table focused on the health-related effects of OPN leaves (line 229).
- Characteristics of OPN fruit: This part also seems extremely complicated, and the literature studied should be better tabulated with the citations.
We agree with Reviewer 1. Therefore, all the citations were thoroughly verified. Furthermore, a new table containing their chemical composition was inserted in this new version of the manuscript to facilitate the comprehension of the OPN fruits' characteristics (line 321).
The conclusion part is well written, and, in the end, the future studies are directed.
Reviewer 2 Report
- The language and organization of this manuscript are not acceptable in its present form. The manuscript can be reconsidered after a substantial revision. My individual comment is listed below.
- Chemical structure of a high arabinogalactan content, characterized as a bi-opolymer composed of the main chain of (1→4) β-D-galactopyranose with branches of galactose, arabinose, rhamnose, and galacturonic acid must be mentioned
- Line 181 chemical formula of ammonium sulfate is (NH4)2SO4
- Line 221&222 non-polar solvents, such as ethanol, acetone, methanol, and hexane. Is methanol and ethanol non-polar?
- You haven't mentioned the antimicrobial activity of OPN leaf extract
- Lines 164& 165 make a table containing the chemical fractionation of amino acids in OPN leaves
Author Response
The language and organization of this manuscript are not acceptable in its present form. The manuscript can be reconsidered after a substantial revision. My individual comment is listed below.
This new version of the manuscript was modified according to the comments of the Reviewers. Eventual language errors were corrected, new scientific information was added, all the citations were carefully revised, and two new tables were inserted. Consequently, we believe that this new version of the manuscript fits the high scientific standards that are required to publish an article in Foods.
- Chemical structure of a high arabinogalactan content, characterized as a bi-opolymer composed of the main chain of (1→4) β-D-galactopyranose with branches of galactose, arabinose, rhamnose, and galacturonic acid must be mentioned.
The description of the chemical structure of arabinogalactan was improved in this new version. The main linkage position of galactose, arabinose, rhamnose, and galacturonic acid was highlighted, and the average molar mass of the arabinogalactan was included (lines 139 – 145).
- Line 181 chemical formula of ammonium sulfate is (NH4)2SO4.
It was corrected in this version (line 183).
- Line 221&222 non-polar solvents, such as ethanol, acetone, methanol, and hexane. Is methanol and ethanol non-polar?
It was a mistake. The expression “non-polar solvents” was replaced by “organic solvents” (line 228).
- You haven't mentioned the antimicrobial activity of OPN leaf extract
Reviewer 02 is right, and the antimicrobial activity of OPN leaves was included in this new version (lines 230 – 233).
- Lines 164& 165 make a table containing the chemical fractionation of amino acids in OPN leaves.
We understand the point of view of Reviewer 02. However, tables containing the amino acid composition of OPN leaves as well as their EAA score were already presented in the two papers we cited in this part of the text, i.e., Takeiti et al. (2009) and Silveira et al. (2020). Therefore, we believe we will be repeating already published data if we insert a table containing the amino acid composition of OPN leaves.
Round 2
Reviewer 2 Report
There is no comment.